

# Cloudiness retrieved from All-Sky camera and MSG satellite over Reunion Island and Antananarivo Madagascar

Jean-Marcel Rivonirina[1,2], Thierry Portafaix[1], Solofoarisoa Rakotoniaina[2], Béatrice Morel[3], Chao Tang[3], Kévin Lamy[1,4], Marie Lothon[5], Tom Toulouse[6], Olivier Liandrat[6], Solofo Rakotondraompiana[2], Hassan Bencherif[1]

[1]LACy, Laboratoire de l'Atmosphère et des Cyclones (UMR8105, CNRS, Université de La Réunion, Météo-France), Saint-Denis, La Réunion, France
[2]IOGA, Institut et Observatoire de Géophysique d'Antananarivo (Université d'Antananarivo, Madagascar), Antananarivo 101, Madagascar
[3]ENERGY-Lab, Université de La Réunion, France
[4]OSU - Réunion, Observatoire des Sciences de l'Univers - La Réunion (UAR 3365, Université de La Réunion, CNRS, Météo-France), Saint-Denis, La Réunion, France
[5]Laboratoire d'Aérologie, Université de Toulouse, CNRS, UPS, Toulouse, France
[6]Reuniwatt, 14 rue de la Guadeloupe 97490 Sainte Clotilde de La Réunion, France

*Correspondence*: Jean-Marcel Rivonirina (jean-marcel.rivonrina@univ-reunion.fr)

**Abstract.** To gain a deeper understanding of cloud variability over the Southwest Indian Ocean (SWIO) region, various measurement techniques can be used. We have used ground-based observations using all-sky cameras from the UV-Indien network provided by the Reuniwatt company, which capture sky images in the visible spectrum. Two algorithms, namely Elifan and Reuniwatt, were applied to analyze the camera images. Despite differences in the methodologies employed by each algorithm, we have found strong agreement between them, with a Bias of -5.48%, a Root Mean Square Error (RMSE) of 6.48%, and a correlation coefficient (r) of 0.99 for Saint-Denis. Ground-based measurements over the SWIO are insufficient, and the vast ocean coverage in this region makes spatial observation important for gathering information. Here we have used cloud products from the Meteosat Second Generation (MSG) satellites. The quality of the classification algorithm employed for camera and satellite plays a significant role in cloud analysis. Comparing camera and satellite observations is essential to ensure the complementarity of each measurement. We observed good consistency between the ground-based camera and satellite measurements (Bias=2.64%, RMSE=21.43%, and r=0.87) with Elifan, and (Bias=6.79%, RMSE=25.70%, and r=0.82) with Reuniwatt for the Saint-Denis site; and (Bias=6.48%, RMSE=28.63%, and r=0.78) for Antananarivo. In Antananarivo, during the dry season, heavy cloud cover (~50%) is observed in the morning, gradually dissipating as the day progresses. Conversely, in the wet season, cloud cover varies between approximately 30% and 60% from December to April, with weaker cloudiness around noon in October and November. As for Saint-Denis, the morning skies are generally clear but become increasingly overcast throughout the day, reaching up to 80% cloud cover during the wet season and 60% during the dry season.



## 1 Introduction

Clouds are primarily composed of water droplets and suspended ice crystals in the atmosphere, formed through the evaporation
of seawater, rivers, and lakes, as well as through plant evapotranspiration, with forests playing a crucial role in increasing
cloud cover, particularly in low altitudes (Duveiller et al., 2021). Aerosols are essential for cloud formation as they act as cloud
condensation nuclei (Ekström et al., 2010). Generally, clouds cover more than half of the Earth's surface. Clouds contribute
significantly to the energy balance at the surface, playing crucial roles in the hydrological cycle and climate change, while also
acting as a natural filter for ultraviolet radiation (UVR). However, clouds can sometimes increase UVR through multiple
scattering phenomena (Sabburg, 2003; Brogniez et al., 2016). They pose challenges to climatic modeling due to their rapid
temporal and spatial variability, and their properties which change with altitude, making them a complex parameter to study
in the atmosphere. The Southwest Indian Ocean (SWIO) region is predominantly covered by the ocean, the interaction between
the ocean and atmosphere in this area is crucial for cloud formation. Our study sites located near the equatorial region, receive
abundant solar radiation, which serves as the primary source of cloud formation. In this region, the wet season extends around
October to April, a period characterized by the formation of tropical cyclones and the highest levels of cloud cover. During
this season, cloud formation is primarily influenced by the Inter-Tropical Convergence Zone (ITCZ) (Vérèmes et al., 2019).
However, cloud cover is low during the dry season. Urban pollution and smoke from fires can reduce cloud formation by
absorbing solar radiation necessary for surface water evaporation. Research has shown that they can reduce cloud formation
to 38% under clear conditions and to 0% under high smoke concentrations (Koren  et al., 2004). Reunion Island presents an
active volcanic site that can influence cloud formation within the region. Moreover, the transport of plumes resulting from
biomass burning in the African region may contribute to a reduction in cloud formation in the SWIO basin.

Cloud data are required in various applications, such as estimating solar energy production (Boudreault et al., 2019), assessing
the influence of UV radiation (Calbó et al., 2005), and modeling climate change (Roebeling et al., 2013). Both active and
passive remote sensing instruments can be used to estimate cloud parameters. Lidar and Radar systems generally provide
vertical distributions of clouds (Vérèmes et al., 2019). Two additional methods (all-sky camera and satellite) were used in this
study to estimate cloud fraction. Satellites are widely used for monitoring cloud evolution with excellent spatial coverage, and
provide information on a large range of temporal scales. Over recent decades advancements in satellite technology have led to
improvements in spatial and temporal resolution, enhancing their capability to provide information across various regions
globally. Satellite data are generally freely accessible, allowing for validation by researchers and significantly contributing to
the understanding of climate change in different regions. Ground-based techniques employing cameras can continuously
monitor cloud evolution with good accuracy, and can directly compared with human observations. They can be used to
calibrate future satellite mission. Both camera and satellite observations techniques are complementary. Various satellite
sensors can be utilized to identify cloud parameters such as cloud type or cloud fraction. One of the most well-known sensors
is MODIS (Moderate Resolution Imaging Spectroradiometer), which belongs to NASA (National Aeronautics and Space
Administration) and is deployed on two satellites: Aqua and Terra (Ning and Wang, 2015). Launched in 2000 and 2002,



MODIS features include 36 spectral bands covering visible to infrared wavelengths, and are dedicated to various topics related to land, ocean, and atmosphere. MODIS provides cloud fraction MOD06 (Terra) and MYD06 (Aqua) products with a spatial resolution of 5 km (Nikumbh et al., 2019), offering two diurnal images over the Indian Ocean. However, geostationary satellites can offer data with high temporal resolution to monitor cloud variability. In this study, we have utilized satellites MSG

(Meteosat Second Generation) operated by Eumetsat (European Organisation for the Exploitation of Meteorological Satellites). MSG provide images every 15 mn with 3 km resolution. The ground-based measurements utilizing all-sky cameras that provide images on which an image processing algorithm is applied to classify images and estimate cloud cover. Various image processing algorithms can be used, such as those based on pixel values and classified images using thresholding criteria (Long et al., 2006), or object-oriented algorithms utilizing texture features (Liu et al., 2015). Recently, observation stations have been

installed in the Indian Ocean by the UV-Indien network (Lamy et al., 2021) to monitor long-term variations in UV radiation and cloud cover, areas where limited studies have been conducted. Due to the predominant ocean coverage in the Indian Ocean region, accessing surface observations is challenging. The installation of the UV-Indien observation station network aims to address this lack of observational instruments in the region. This network comprises 10 observation stations located across various regions of the SWIO, but here we have focused on two sites Antananarivo Madagascar, and Saint-Denis of Reunion

Island. In this study, we evaluate the performance of two different algorithms used with the camera: a proprietary industrial algorithm developed by the Reuniwatt company, which processes images taken by an all-sky camera (Liandrat et al., 2017) for measuring cloud fraction, and another algorithm named Elifan, originally developed by CNRS (Centre National de la Recherche Scientifique) and currently used by the ACTRIS-France (Aerosol, Cloud and Trace Gases Research Infrastructure - France) community (Lothon et al., 2019). The latter was adapted for the case of Reunion Island.

The main objective of this study was to examine cloudiness properties in two locations in the Indian Ocean: Antananarivo Madagascar, and Saint-Denis Reunion.

This paper is structured as follows: the data and methodology section, which includes the study area, camera presentation, and algorithm description for obtaining nebulosity data; the results section; and finally, the discussion and conclusion.

## 2 Data and methodology

### 2.1 Study area

As mentioned previously, we are working on two different sites in SWIO; the first station is located in the city of Saint-Denis in Reunion Island a French department, where an all-sky camera is positioned on the roof of the Faculty of Sciences at Reunion University (Fig. 1). This university is located on the northern coast of the island, at an altitude of 70 meters above mean sea level, with longitude and latitude coordinates of (55.485° E, 20.902° S). To the north of the station lies the ocean, while to the

south, there are towering mountains. The microclimate resulting from the topography of Reunion Island presents a particularly complex subject for study. The presence of two highest summits in the island, Piton des Neiges at 3071 m above sea level, and the active Piton de La Fournaise volcano at 2560 m above sea level, influences atmospheric circulations in the SWIO (Mialhe



et al., 2020). The location of Saint-Denis provides a significant advantage for studying land and sea breezes. One notable advantage of this site is its multi-instrumentalization dedicated to atmospheric measurements (Baray et al., 2013; Durand et

al., 2021). Our study contributes to filling the gap in our understanding of this site, allowing for inter-comparison of ground-based instruments with spatial observations. The second site is located in Antananarivo in the highlands of Madagascar, where the camera is situated at a top of a pillar on the hill of Ambohidempona, within the premises of the Institute and Observatory of Geophysics of Antananarivo (IOGA) at the University of Antananarivo. The coordinates of this site are (47.565° E, 18.916° S) at an elevation of 1370 m. These both stations were the first of the UV-Indien network, established in 2019 (Table 1).

Limited documentation is available for Antananarivo, while numerous studies have been conducted on Reunion Island.

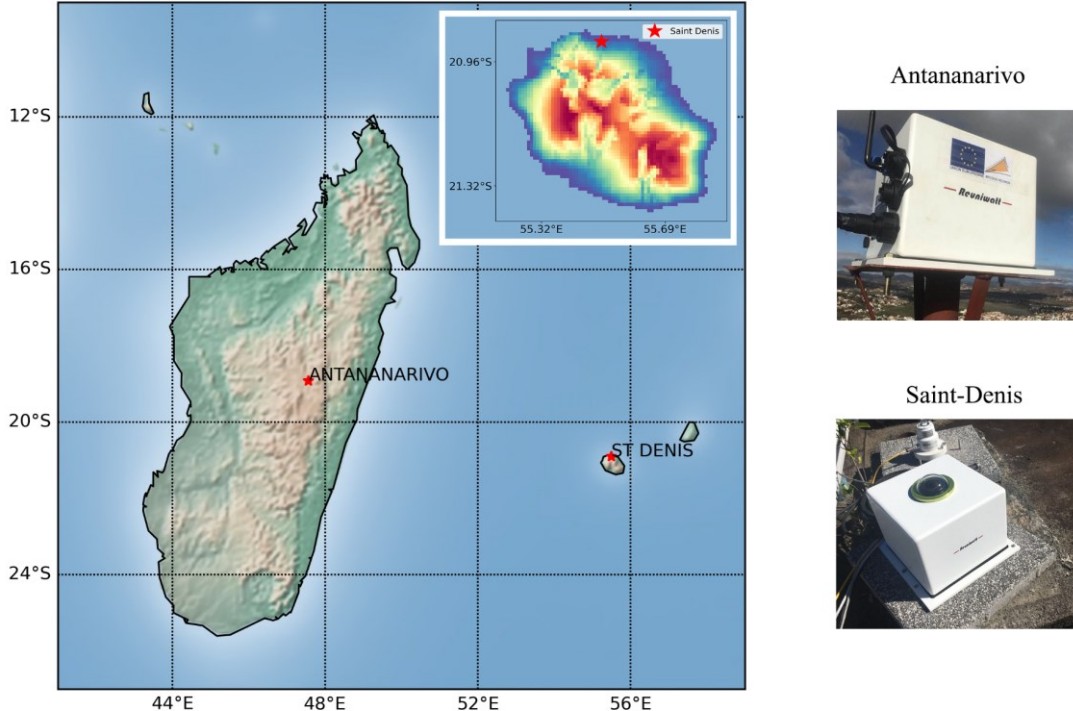

**Figure 1.** Cameras location, Saint-Denis - Reunion and Antananarivo - Madagascar.

The data used in this study are presented in Table 1 below. For uniformity of data, we have focused our comparison (Sect. 3.1) between September 2019 and December 2021, and between September 2019 and Jun 2022 for cloud variability (Sect. 3.2). An extended period can be applied to establish a climatology of cloud variability when we have enough data.

**Table 1.** Camera and satellite data set used in the process.

| Location | Parameters | Instrument | Algorithm | Date | Resolution | Frequency | Reference |
|---|---|---|---|---|---|---|---|
| Saint-Denis - Réunion Latitude: 20.902°S | Cloud Fraction | Sky Cam Vision | Reuniwatt | 13/09/2019 - Present | 2048x2048 pixels | 30 s | Cadet et al., 2020 |
| | | | Elifan | 01/09/2019 - 20/12/2021 | | | Lothon et al., 2019 |



| | | | | | | | |
|---|---|---|---|---|---|---|---|
| Longitude: 55.485°E | Cloud Mask | MSG/SEVIRI | - | 01/09/2019 - 30/06/2022 | 3 km | 15 min | Derrien and Le Gléau, 2005 |
| | Cloud Type | MSG/SEVIRI | - | 01/09/2017 - 30/06/2022 | | | |
| Antananarivo-Madagascar Latitude: 18.916°S Longitude: 47.565°E | Cloud Fraction | Sky Cam Vision | Reuniwatt | 06/2019 - Present | 2048x2048 pixels | 30 s | Cadet et al., 2020 |
| | Cloud Mask | MSG/SEVIRI | - | 01/09/2019 - 30/06/2022 | 3 km | 15 min | Derrien and Le Gléau, 2005 |
| | Cloud Type | MSG/SEVIRI | - | 01/09/2017 - 30/06/2022 | | | |

## 2.2 Camera

To detect Cloud Fraction (CF), we utilized the all-sky imager "Sky Cam Vision", a commercial camera manufactured by Reuniwatt company (https://reuniwatt.com/en/2019/11/11/observe-the-sky-with-our-sky-imagers/, last accessed on 26/10/2023). The camera is directed towards the zenith of the designated site to capture images of cloud cover variability from the ground within the visible range. It is outfitted with a "fish-eye" lens housed in a small glass dome to shield it from rain and weather fluctuations. The camera produces hemispherical images in high definition range (HDR) with a field of view of 360°

x 180° around the site where the camera is positioned. The resolution of the images is 2048 × 2048 pixels, acquired at intervals of 30 seconds that can be adjusted through the user interface. The algorithms applied on the camera images, namely Reuniwatt and Elifan, are based on the red over blue ratio (R/B) obtained from images, hereafter referred to as RBR. They are detailed in Sect. 2.3 and 2.4.

## 2.3 Reuniwatt algorithm

Reuniwatt performs image segmentation using a Random Forest algorithm (Breiman, 2001) to classify pixels into different categories. The Random Forest learning is conducted on several color spaces using thousands of pixels from sky cam vision images that have been manually labeled. The pixels were labeled into 4 categories: clear sky, thin cloud, thick cloud, and sun. This process results in an image containing these 4 labels, which allows for two types of segmentation. One segmentation identifies only thick clouds, while the other identifies all clouds. In all cases, the sun pixels are considered as clear sky pixels

in the segmented image. In this study, the cloud fraction used has been computed on the segmentation of all clouds. The cloud fraction is then calculated using a weighted average based on the angles of pixels in a geometrically calibrated image. This method is described in Long et al. (2006). The Random Forest algorithm performs pixel classification utilizing texture and shape information from the images, providing an improved segmentation compared to thresholding alone. This allows Reuniwatt to obtain cloud fraction outputs with high accuracy.

## 2.4 Elifan algorithm

Elifan is an image-processing algorithm initially developed by CNRS in 2013 and is currently used by the ACTRIS-France community (Lothon et al., 2019). It allows us to get daily nebulosity or cloud fraction index from all-sky cameras and is





operational at various measurement sites of ACTRIS-France. The data is processed and centralized at the AERIS data center to ensure homogeneity. We have adapted the Elifan algorithm on two new sites of ACTRIS-France located in Reunion Island

on which we have a Reuniwatt camera, at the University of Moufia Saint-Denis and Maido Observatory (Baray et al., 2013). Two different classification methods can be employed: one based on pixel values (pixel-oriented) and the other based on the texture or shape of the image (object-oriented). In our case, the algorithm utilizes classification by the pixel value. Two thresholding methods, namely absolute and differential, detailed in Sect. 2.4.3 and 2.4.5, respectively, are applied to the RBR to distinguish clouds from the blue sky (Lothon et al., 2019). However, various pre-processing steps, as outlined below, are

first applied to the images. Each thresholding method has its strengths: absolute thresholding yields better estimation when the sun is obscured by clouds, whereas differential thresholding is generally good in other scenarios (Lothon et al., 2019). All the processing steps of Elifan are detailed further in Lothon et al. (2019). Here, we will only present the steps that have been adapted for the case of Reunion as follows:

- Cropping image and creating object masks
- Generating solar masks
- Creating a library of blue sky images
- Defining absolute and differential threshold values

### 2.4.1 Cropping image and object mask creation

To crop the observed image, we have selected a 70° radius angle around the zenith, which is equivalent to the angle chosen in

the Reuniwatt algorithm. This is done to exclude areas near the horizon where pixels show significant distortion or deformation, thus making interpretation challenging. Another rationale for this choice is to avoid including objects such as buildings that may appear in various observation sites. The white circle in Fig. 2 delineates the boundary, indicating that only the image inside the circle will be considered for the subsequent processing steps.

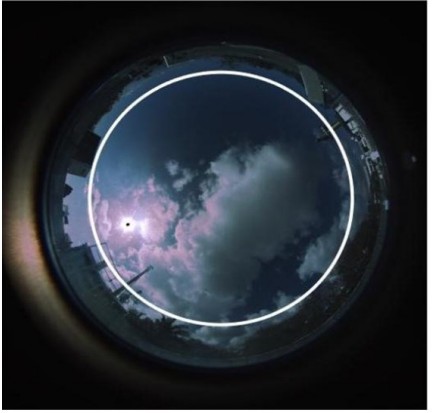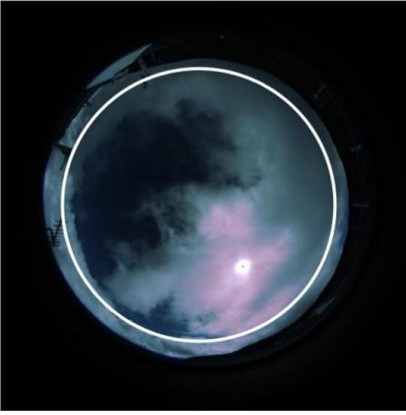

**Figure 2.** Cropped area delimitation, Saint Denis (left) and Maïdo (right)





Depending on the site, certain elements with moving parts may not be completely eliminated during the cropping step, as
observed at the Moufia Saint-Denis site (Fig. 4). To address this issue, we have defined an additional object mask, depicted in
black (Fig. 4a).

**2.4.2 Solar mask**

The creation of the mask is necessary to exclude the area surrounding the sun, which tends to be oversaturated in the image
and can lead to overestimation of results. To address this issue, a dynamic mask that varies according to solar angles (zenith
and azimuth) is required. Initially, we have conducted the process without utilizing a sun mask by selecting images from a day
when the sun is unobscured by clouds. The goal was to obtain samples of solar positions to establish Eq. (1) below. We have
employed cubic regression, which provides the most accurate representation of the sun's trajectory in the image. This method
draws inspiration from the approach used by Lothon et al. (2019), where the position of the solar mask adjusts according to
both input variables: the zenith angle (α) and the solar azimuth angle (θ).

$$\begin{cases} I_S = A1 * \beta^3 + B1 * \beta^2 + C1 * \beta + D1 \\ J_S = A2 * I_S{}^3 + B2 * I_S{}^2 + C2 * I_S + D2 \end{cases} \text{ where } \beta = \sin\left(\frac{\alpha}{2}\right)\sin\left(\frac{\theta}{2}\right) \qquad (1)$$

$I_S$ represents the abscissa and $J_S$ represents the ordinates of pixels in the image. A1, B1, C1, D1, A2, B2, C2, and D2 are the
coefficients obtained from samples extracted from the image processing of a day without a mask, where the area around the
sun in the image is not obscured by clouds. Once we have obtained the equation, the coefficients D1 and D2 can be manually
adjusted to correct deviations. We have taken an example of 29/08/2020 (Fig. 3), a date on which the sky is almost completely
clear throughout the day.



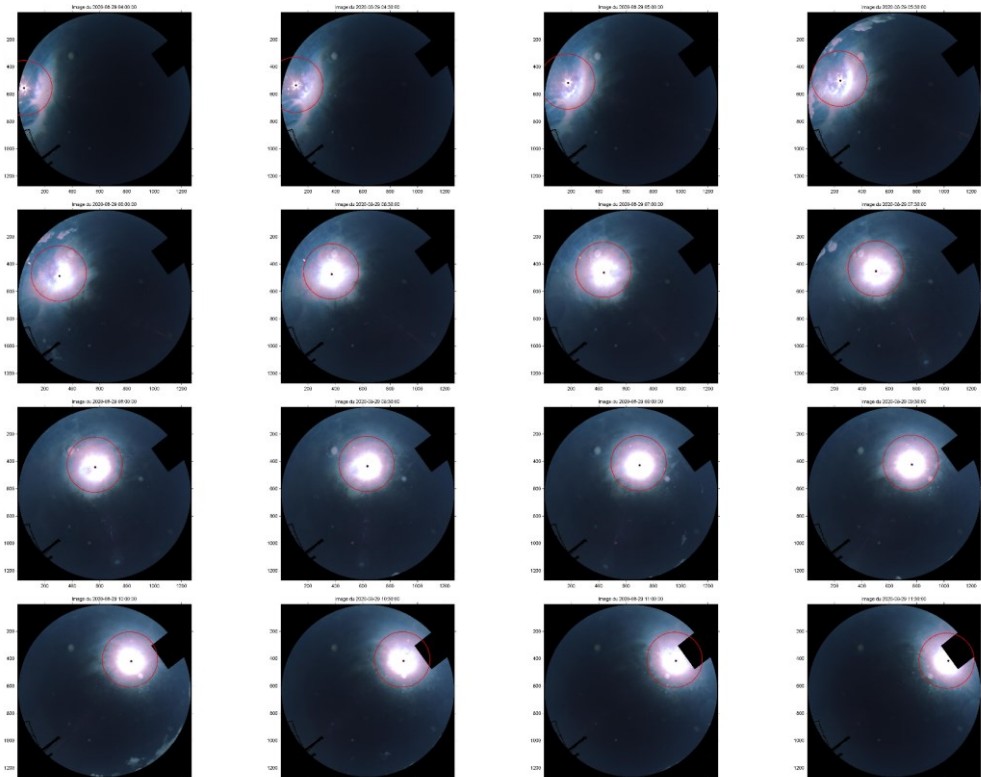

**Figure 3.** Simulated trajectory of the sun on 29/08/2020, every 30 mn from 4:00 to 11:30 (UTC), indicated by the red circle.

### 2.4.3 Absolute thresholding

The classification is determined by applying different thresholding definitions to the RBR. Two sets of values have been defined: one set consists of values greater than the threshold denoted by $T_{Cloud}$, and the other set consists of values less than

the threshold $T_{Blue\ Sky}$. These values can be found in Table 2. From these threshold definitions applied to the RBR, three distinct pixel classes are derived: Cloud, Blue Sky, and Uncertain. The thresholds are defined as follows: pixels with RBR values less than $T_{Blue\ Sky}$ are classified as "Blue Sky", pixels with RBR values greater than $T_{Cloud}$ are classified as "Cloud", and pixels with RBR values between $T_{Blue\ Sky}$ and $T_{Cloud}$ are categorized as "Uncertain".

-    Blue Sky   (RBR $\leq T_{Blue\ Sky}$)

-    Uncertain ($T_{Blue\ Sky} <$ RBR $\leq T_{Cloud}$)

-    Cloud (RBR $> T_{Cloud}$)

### 2.4.4 Blue Sky Library Creation

The establishment of the blue sky image library is essential for conducting the process of differential thresholding. To determine whether an image is clear or cloudless, we are using the condition outlined by the following relation (2).

$\min(RBR) < 0.5 \ \& \ \max(RBR) > 0.6 \ \& \ count(Incertain) < 6\%$      (2)



If an image satisfies this condition, it will bypass the algorithmic classification process and be directly added to the clear sky library. The constant parameters may differ depending on the type of camera used and the specific site being studied, as aerosol levels at each site can affect the quality of image classification (Lothon et al., 2019). However, in our case, aerosol is not taken into account in the algorithm. Threshold values can be visually determined from the multi-color image (Fig. 4). The values specified here in Eq. (2) correspond to those of the Moufia-Saint-Denis station.

### 2.4.5 Differential thresholding

In the image, multiple solar reflections on the plexiglass protecting the fisheye lens are sometimes present. Additionally, saturations that could not be eliminated by the sun mask occur. To minimize these effects, we have utilized a clear sky image library. The principle is the same as that of the absolute thresholding method using the RBR, but it is only applicable when a clear sky image with the same solar angle (azimuth and zenith) as the processed image is available in the library. This method is inspired from Ghonima et al. (2012). The concept is straightforward: subtract the RBR of the processed image from that of the reference clear sky image with the same solar angle in the library.

- Bleu Sky ($RBR - RBR_{Lib} \leq T_{Blue\,Sky}$)
- Uncertain ($T_{Bleu\,Sky} < RBR - RBR_{lib} \leq T_{Cloud}$)
- Cloud ($RBR - RBR_{Lib} > T_{Cloud}$)

The threshold values defined in differential methods are presented in Table 2 below. Depending on the camera model and its configuration, as well as the characteristics of aerosols or haze variability at each station, these threshold values may vary from one site to another (Lothon et al., 2019). The differences between absolute and differential methods are not significant, as demonstrated in Fig. 4. This is why we just utilize absolute thresholding for comparison in Sect. 3.1.

**Table 2.** Threshold value chosen for Saint-Denis and Maïdo sites.

| Site | Absolute | | Differential | |
| :---: | :---: | :---: | :---: | :---: |
| | $T_{Blue\,Sky}$ | $T_{Cloud}$ | $Tdiff_{Bleu\,Sky}$ | $Tdiff_{Cloud}$ |
| Saint-Denis | 0.55 | 0.6 | 0.05 | 0.1 |
| Maïdo | 0.5 | 0.55 | 0.1 | 0.15 |



**Figure 4.** Intermediate image processing results at Moufia Saint-Denis - 30/09/2019 at 09:54:00 (UTC). (a) Cropped image containing the solar mask represented by the red circle, and the object mask in black. (b) Multi-color image processed using the absolute thresholding method. (c) Tricolor image processed with the absolute thresholding method. (d) Multi-color image processed using the differential thresholding method. (e) Tricolor image processed using the differential thresholding method.



## 2.5 MSG / SEVIRI

Meteosat Second Generation (MSG) is a meteorological satellite mission operated by Eumetsat. Four satellites positioned at
210   an altitude of 36000 km have been synchronized to provide meteorological data. They provide images of the full disk with
dimensions of 3712 x 3712 pixels, covering Europe, Africa, and parts of the Indian Ocean since 2002 (Werkmeister et al.,
2015). Operating in a geostationary orbit, the MSG mission focuses on observing various parameters including clouds, land,
and ocean surfaces. The MSG mission concluded in 2022 and was recently replaced by the Meteosat Third Generation (MTG-
I1) satellite. The latter offers significant improvements in spatial and temporal resolution by providing images every 10 mn
215   with three resolutions: 0.5, 1, and 2 km, corresponding to different wavelength bands. MSG employs the Spinning Enhanced
Visible and Infrared Imager (SEVIRI) sensor, which captures images in 12 spectral bands ranging from visible to infrared.
These include three visible channels (0.6, 0.8, and 1.6 μm), eight thermal infrared channels (3.9, 6.2, 7.3, 8.7, 9.7, 10.8, 12,
and 13.4 μm), and one high-resolution broadband visible channel (Taravat et al., 2015). The spatial resolution varies from 1 to
3 km depending on the bands or wavelength range. In 2016, the MSG satellites were repositioned at a longitude of 41.5° E to
center images over the Indian Ocean, continuing the service previously provided by Meteosat 7. MSG took images every 15
mn, offering significant advantages by providing day and night observations with infrared bands, resulting in a total of 96
images per day. This study utilizes the Cloud Mask (CLM), a product obtained through multispectral threshold tests that
classify pixels into different categories (Bley and Deneke, 2013). Further details of the algorithm to derive CLM are provided
in Derrien and Le Gléau (2005). The pixel values in CLM images are defined in Table 3 below. CLM has a spatial resolution
of 3 km × 3 km and is available upon request and free of charge from the Eumetsat website https://www.eumetsat.int/ (last
accessed on 26/10/2023).

**Table 3.** Pixels values in Cloud Mask.

| Parameter | Value | Description |
|---|---|---|
| CLM | 0 | Clear sky over water |
| | 1 | Clear sky over land |
| | 2 | Cloud |
| | 3 | No data |

Cloud Type (CT) data can be retrieved from the ICARE (Cloud-Aerosol-Water-Radiation Interactions) data center platform at
https://www.icare.univ-lille.fr/asd-content/extract/subset/ordergeo (last accessed on 26/10/2023). Due to redundancy and the
absence of certain classes in this region, reclassification is necessary. To facilitate data interpretation, we have redefined the
pixel values in CT, initially defined by SAFNWC (Satellite Application Facility for supporting NoWCasting and very short-
range forecasting), into six different classes: cloud-free, low clouds, medium clouds, high opaque clouds, fractional clouds,
and high semi-transparent clouds (refer to Fig. 5). A similar study involving the reclassification of cloud-type products has
been previously conducted by Philippon et al. (2016).





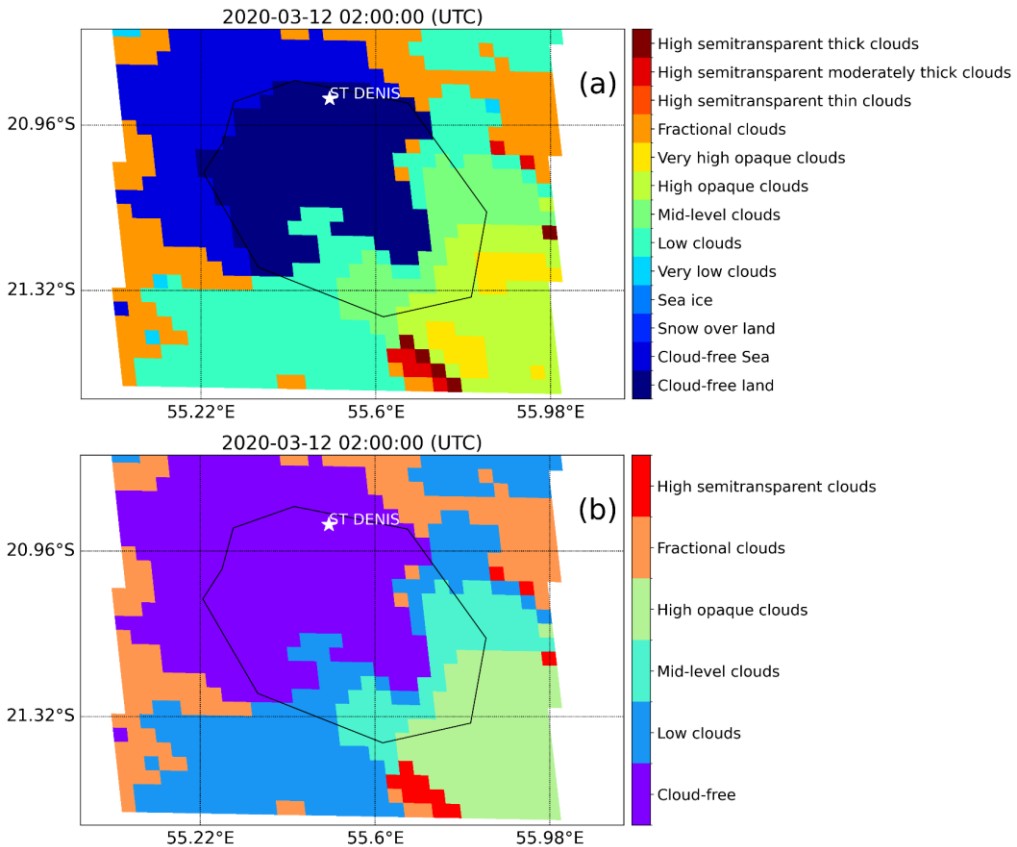

**Figure 5.** Initial classification of cloud type of MSG/SEVIRI (a). Reclassification of pixels values (b).

**2.6 Statistical parameters**

To compare each measurement, the following statistical parameters were utilized. Eq. (3) computes the bias, enabling us to estimate differences between both measurements. The variability of the measurement can be determined using the Root Mean Square Error (RMSE), as shown in Eq. (4). The performance of each comparison can be evaluated by the correlation coefficient (r), as expressed in Eq. (5).

$$Bias = \frac{1}{N}\sum_{i=1}^{N}(x_i - y_i) \qquad (3)$$


$$RMSE = \sqrt{\frac{1}{N}\sum_{i=1}^{N}(x_i - y_i)^2} \qquad (4)$$

$$r = \frac{\sum_{i=1}^{N}(x_i - \overline{x_i}) \times (y_i - \overline{y_i})}{\sqrt{\sum_{i=1}^{N}(x_i - \overline{x_i})^2 \times \sum_{i=1}^{N}(y_i - \overline{y_i})^2}} \qquad (5)$$





## 3 Results

### 3.1 Comparison

#### 3.1.1 Elifan against Reuniwatt

Table 4 compares the Elifan and Reuniwatt algorithms for cloud fraction retrieval. It is important to note that direct comparison is challenging because they analyze different sky regions. Reuniwatt considers the entire sky image, while Elifan masks out the area surrounding the sun to avoid saturation. This masking explains the observed difference, where Reuniwatt generally yields slightly higher cloud fractions than Elifan. Despite this difference, both algorithms demonstrate good agreement with a low Root Mean Square Error (RMSE) of 6.48% and a high correlation coefficient (r=0.99). The small offset is likely due to

Elifan's exclusion of the solar zone, which is often saturated and can lead to overestimation. This difference is more pronounced around noon when solar irradiance is most impactful. Therefore, both algorithms offer comparable performance for cloud fraction calculations. Since Elifan couldn't be applied at the Antananarivo site due to data management reasons, Reuniwatt can be confidently used without compromising accuracy.

**Table 4.** Statistical parameters (root mean square error "RMSE", correlation coefficient "r", and number of observation "N") to compare each measurement.

| Site | Comparison | Bias (%) | RMSE (%) | r | N |
|------|------------|----------|----------|---|---|
| | Elifan-Reuniwatt | -5.48 | 6.48 | 0.99 | 23 244 |
| Saint-Denis – Réunion | Elifan - MSG | 2.64 | 21.43 | 0.87 | 20 925 |
| | Reuniwatt - MSG | 6.79 | 25.70 | 0.82 | 25 807 |
| Antananarivo – Madagascar | Reuniwatt - MSG | 6.48 | 28.63 | 0.78 | 27 478 |

#### 3.1.2 Camera against Satellite

Each measurement from ground or space-based sources has its advantages and weaknesses. Camera images operate within a visible wavelength range, which constitutes their limitations. Cameras often struggle to accurately estimate cloudiness, particularly around sunrise and sunset (Lothon et al., 2019). The uncertainty of cloud masks from MSG satellites is primarily related to the spatial and temporal variability of surface reflectance, caused by changes in atmospheric aerosols or vegetation. Thin cirrus clouds are sometimes undetected by satellites because their spectral signatures are similar to clear skies (Taravat

et al., 2015). The cloud fraction (CF) at each site was obtained from CLM images using the following procedure. We have considered the pixels within a square window of 3×3 pixels centered on the nearest point to the station. The number of cloud pixels included in the window was calculated and divided by the total number of pixels within the same window. Pixel values are represented by points, as depicted over Reunion Island in Fig. 6 below, where dark blue dots represent cloud pixels.



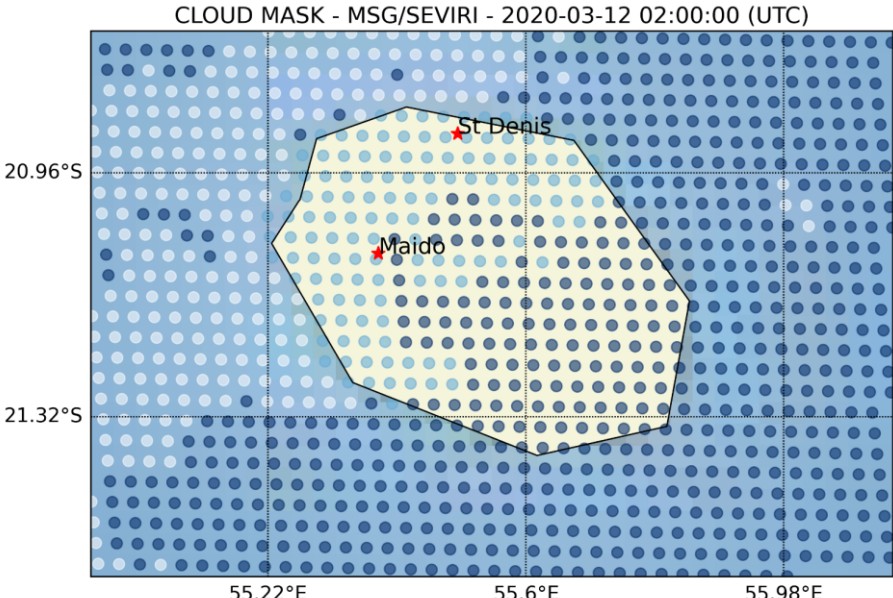

**Figure 6.** Pixels extracted from Cloud Mask of MSG/SEVIRI over Reunion Island where cloud pixels are represented by dark blue dots.

The choice of pixel window size to calculate cloud fraction from the Cloud Mask of MSG/SEVIRI can significantly impact

the quality of comparison. Different window sizes, such as 3×3, 5×5, 7×7, or more, can be considered. We have opted for a dimension of 3×3, which closely aligns with camera observations and is less influenced by topographical variability. Increasing the window size tends to amplify the differences between camera and satellite observations. We have compared the outputs of the Elifan and Reuniwatt algorithms with MSG data (refer to Table 4). The data used to calculate bias corresponded to satellite overpasses with a 15 mn interval between September 2019 and December 2021. We have selected a time range between 8:30

and 16:30 (LT) to avoid the camera's limitations during sunset and sunrise. Despite differences in the dimensions of the camera and satellite images, we found good consistency between the data. Elifan exhibited the lowest bias (Bias=2.64%), RMSE=21.43%, and high correlation (r=0.87), Reuniwatt showed slight differences (Bias=6.79%, RMSE=25.70%, and r=0.82) for the Saint-Denis site and (Bias=6.48%, RMSE=28.63%, and r=0.78) for Antananarivo. Statistical values for Saint-Denis generally outperform those for Antananarivo in terms of consistency between MSG and Reuniwatt. This disparity is

attributed to differences in environmental and atmospheric properties at each site. Satellites may struggle to identify low cloud cover, occasionally due to the presence of haze at the study site. Cloud fraction bias is primarily influenced by image resolution and cloud distribution (Jones et al., 2012), varying between sites and being linked to atmospheric characteristics. Pollution levels differ at each station, particularly in aerosol concentration, which can attenuate solar radiation intensity (Radivojevi et al., 2015). Aerosol levels are higher over Antananarivo compared to Saint-Denis (Lamy et al., 2021), potentially inducing

slight biases between the two sites.



## 3.2 Seasonal cloud variability

Fig. 7 and 8 have been divided into two seasons: the top represents the wet period, and the bottom represents the dry season. Fig. 7 illustrates the seasonal variability of cloudiness at the Antananarivo site as obtained from the camera using the Reuniwatt algorithm and from the MSG/SEVIRI satellite.

At the Antananarivo station, during the dry season, we generally observe heavy cloud cover in the morning, around 50%, which gradually decreases throughout the day. Conversely, during the wet season, cloud cover ranges from around 30% to 60% in the morning, increasing as the day progresses. These trends are consistent across both camera and satellite observations. In the wet season, the lowest coverage is typically observed in October and November, reaching a minimum close to 20%, while the maximum value exceeds 60% in February. There is a slight deviation from December to April. For the Saint-Denis

site (Fig. 8), we observe similar variability between camera and satellite data: the sky exhibits low coverage or is almost clear in the morning, with cloudiness gradually increasing during the day. In the wet season, maximum cloud cover can reach 80%, compared to only 60% during the dry season. The lowest coverage is typically observed in May, below 20%, while the maximum occurs in January. Cloud fraction generally tends to be weaker in the dry season compared to the wet season at both sites. Despite differences in resolution, spatial dimension, and field of view between camera and satellite measurements, we

find good consistency between both datasets. The selected sites exhibit significant differences in geographical and environmental properties. The Antananarivo station, situated at an average elevation of 1370 m, experiences wind patterns along the mountains that influence orographic cloud formation, as further detailed by Rana and Sathiyamoorthy (2018). In contrast, the Saint-Denis site, located at an elevation of 70 m above sea level, experiences various sources of cloud formation, including orography influenced by the local topography of Reunion (Mialhe et al., 2020), evaporation of seawater, and

evapotranspiration due to the island's high vegetation coverage.

The diverse roles that clouds play on Earth's surface underscore the importance of cloud data from the UV-Indien network. Due to the extensive sea surface coverage in some Indian Ocean sites, the spatial resolution of satellites is sometimes inadequate for data analysis, highlighting the importance of ground-based stations.



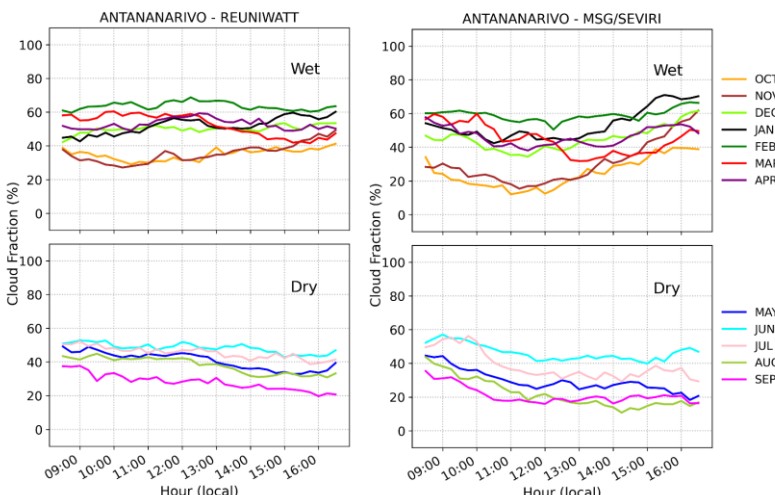

**Figure 7.** Seasonal variability of nebulosity over Antananarivo, Madagascar, obtained from Reuniwatt on the left and MSG/SEVIRI on the right side (September 2019 - Jun 2022). Hour in abscissa represents local time (UTC+3).

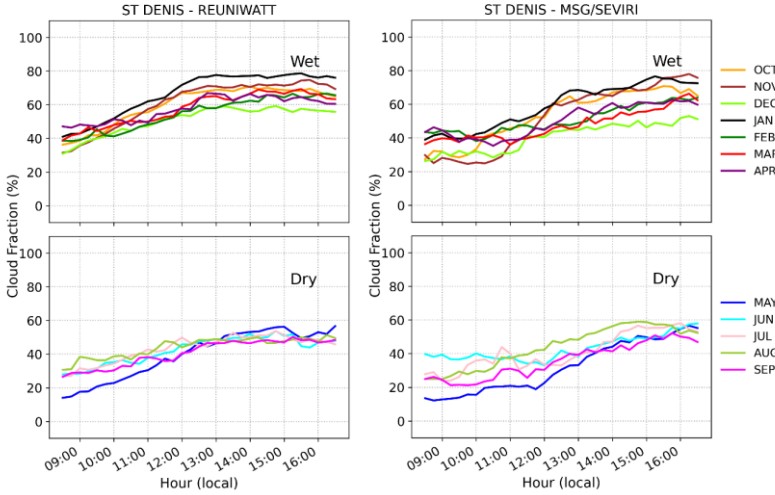

**Figure 8.** Same as Fig. 6, over Saint-Denis Reunion with local time (UTC+4)

### 3.2.1 Cloud frequency over Antananarivo

The occurrence of clouds each month, obtained from the Cloud Type data of MSG/SEVIRI over Antananarivo, is illustrated in Fig. 9. The histograms represent the percentage of all observations, for a total of 47960 observations between September 2019 and June 2022. Notably, more than 46% of the observations depict a clear sky, with maximum monthly values exceeding 50% observed from May to October. Clear skies are associated with significant solar irradiance. Low clouds, typically thick




and dark, significantly reduce ground-level solar radiation, accounting for approximately 21% of observations. High semi-transparent clouds, constituting around 16% of observations, and fractional clouds, representing more than 10%, can intensify solar irradiance. High opaque clouds and middle-level clouds are less frequent, each representing only 3% of observations. The diurnal distribution of sky covers from September 2019 to June 2022 is depicted in Fig. 10, where all types of clouds are observed. Clear skies, one of the most dominant classes, are prevalent throughout the day. High semi-transparent clouds are

more frequent, while low clouds, typically thick, are predominant in the morning and less frequent in the afternoon. High semi-transparent clouds are consistently observed throughout the day, with a frequency of approximately 25%. High opaque clouds are primarily observed towards the beginning of the afternoon. The variation in cloud type during the seasons is strongly influenced by geometric parameters such as Earth-sun distance and solar zenith angle. Clear skies are more frequent during the austral winter compared to summer, and high opaque clouds are absent during austral winter. Middle-level clouds are less

frequent throughout the year. Additionally, the presence of Mandroseza lake near the station can contribute to the cloud type in this site.

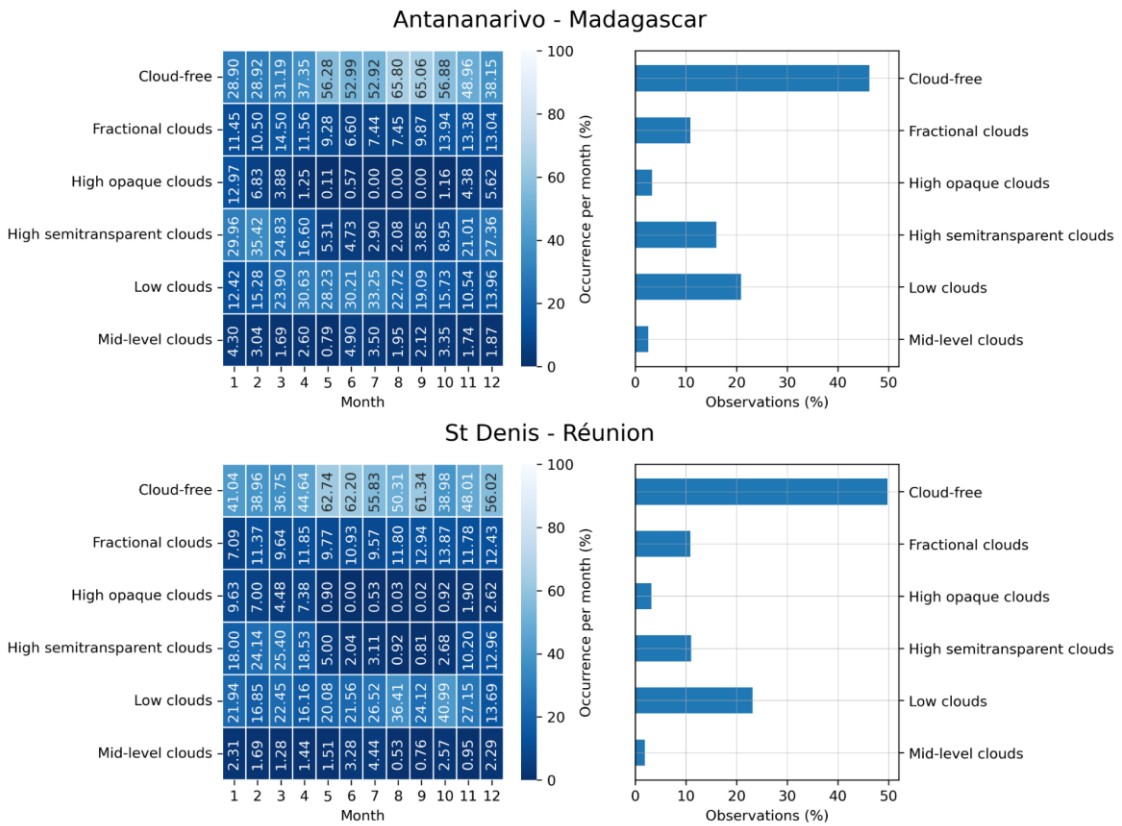

**Figure 9.** Monthly observation of sky from September 2019 to Jun 2022 over Antananarivo and Saint-Denis obtained from Cloud Type of MSG/SEVIRI.



### 3.2.2 Cloud frequency over Saint-Denis

We have conducted an analysis of the monthly occurrence of different cloud classes over Saint-Denis (Fig. 9). Similar to Antananarivo, we observe a strong occurrence of clear skies, representing 50% of observations. The peaks in clear sky

occurrence are evident from May to December, except in October and November, characterized by the strong presence of low clouds. Low clouds are dominant in Saint-Denis, with a frequency of approximately 23% throughout most of the year. High semi-transparent clouds are generally significant (>10%) during the wet season from November to April, while fractional clouds are consistently observed each month at around 11%. High opaque clouds are less visible except during January, February, March, and April, with occurrence frequencies exceeding 4%. Mid-level clouds are less present, with an average

frequency of around 2% each month. The diurnal distribution of cloud types is consistent with cloud fraction observations, particularly low clouds and fractional clouds, which show low occurrence in the morning and increase during the day.

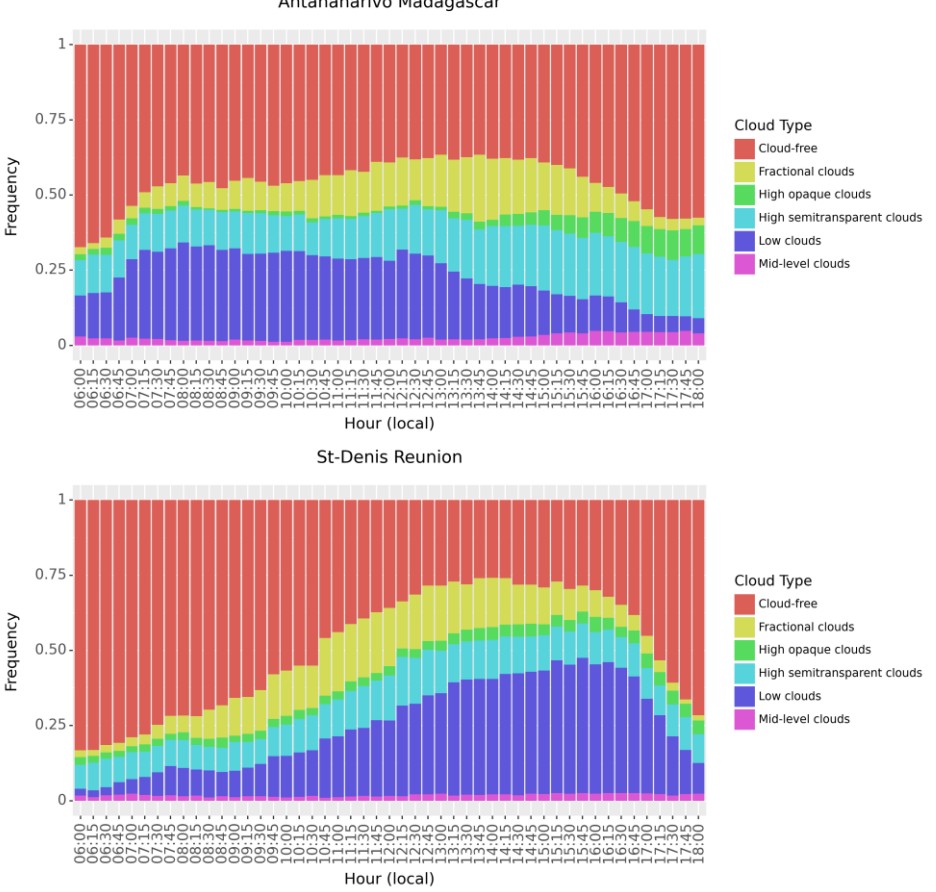

**Figure 10.** Diurnal distribution of cloudiness over Antananarivo, Madagascar, and Saint-Denis Reunion (September 2019 - Jun 2022).



## 4 Discussion and conclusion

Research on cloud variability in the SWIO is currently insufficient. The high spatial and temporal resolution of the data used in this study enhances our understanding of cloud characteristics and variability across the region. Deviations in each measurement were estimated using statistical analysis. Although the methodologies employed by the Elifan and Reuniwatt algorithms differ, they both reliably estimate cloud fraction (Bias=-5.48%, RMSE=6.48%, and r=0.99). Between satellite and camera measurements, significant biases sometimes arise due to the differing spatial dimensions of the measurements. Discrepancies often occur when clouds outside the camera's field of view are captured by the satellite. Satellite estimates tend to overestimate when very high clouds like cirrus are present. Conversely, cameras may overestimate cloudiness when fog is present around the site, making the area more cloud-covered compared with satellite images. Additionally, satellites sometimes fail to identify low-level cloud cover that appears dominant in ground observations (Verma et al., 2018).This study shows us the consistency and inter-complementarity of camera and satellite data. Camera and satellite statistics comparison on Saint-Denis (Bias=6.79%, RMSE=25.70%, and r=0.82) is generally good compared with Antananarivo (Bias=6.48%, RMSE=28.63%, and r=0.78). The accuracy of comparisons can be further improved by using data from the MTG-I1 satellite, which offers better spatial and temporal resolution compared to MSG. Cameras provide more consistent data with visual observations due to their good spatial resolution and high temporal frequency, although ground measurements have limitations; they are local and representative only within a spatial scale of about 5 km. Beyond this, correlations between two ground stations are inconsistent (Kalecinski, 2015). Both ground-based and satellite measurements are complementary; however, the choice depends on the specific area and focus of the study. In specific cases using satellites offers a good perspective, especially for areas that do not have camera installation as in the case of some sites of UV-Indien network. The data collected by both cameras and satellites can be applied in various fields, such as estimating solar power plant production (Rodríguez-Benítez et al., 2021). This study's datasets will also enhance our understanding of clouds effects on UV radiation, particularly for the UV-Indien network. Different cloud types impact solar irradiance differently, underscoring the necessity of cloud classification (Akdemir et al., 2022). Additional global horizontal irradiance (GHI) and diffuse horizontal irradiance (DHI) data provide by IOS-net network composed by 20 stations around SWIO can be used to understand these cloud effects (Morel et al., 2021). Incorporating other satellite data like MODIS, which offers similar spatial resolution to MSG, is a promising approach. Additional observational tools, such as ground-based cloud radar providing vertical cloud distribution or a combination of radar and lidar as used by Vérèmes et al. (2019), can offer more detailed insights into cloud variability over specific sites. Cloud classification errors can occur due to overlapping clouds at different altitudes, with some cloud types being obscured by others. Additional biases were identified when comparing Cloud Type and Cloud Mask products, revealing that fractional clouds sometimes mix with clear skies. The occurrence frequency of clouds reported here aligns with findings by Durand et al. (2021), who noted a maximum occurrence frequency of about 45% for low clouds from 12 to 19h (LT) during the wet season, and a peak of 15% for high clouds from 13 to 18h (LT). The cloudiness variability presented in this study is not fully representative due to the limited depth of our archives, but it provides a useful overview of cloud cover variability at poorly



known sites. This study spans nearly three years, and continuous analysis over a longer period could reveal more about cloud formation and variability. But even with this limited data, we can observe the distinctiveness of cloudiness at each site, Saint-Denis and Antananarivo. This is influenced not only by atmospheric circulation but also by various factors such as environmental and geographic conditions.

*Author contributions.* JMR prepared the manuscript. TP supervised the study. TT wrote Reuniwatt algorithm description. All authors reviewed the manuscript.

*Financial support.* This study is financially supported by MOUV.RE project co-financed by the European Union and the Reunion region. This study is also supported by IEA CNRS project MAGNET (Study of cloud cover in Madagascar, impact
on UV radiation).

*Competing interests.* The authors declare that they have no conflict of interest.

*Acknowledgment.* We would like to thank the UV-Indien project for installation of the network of all-sky camera in Indian
Ocean. We thank Reuniwatt society for their help on technical support during camera installations. Thanks to OSU-Reunion (Observatoire des Sciences de l'Univers - La Réunion) for camera data management. We would like to thank Eumetsat team for sharing MSG cloud mask data, and colleagues at ENERGY-Lab for their help for MSG cloud type data retrieval from ICARE data center. Last not least, we thank French Government for supporting parts of PhD student thanks to Eiffel scholarship.

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
