# Peer review of "Cloudiness retrieved from All-Sky camera and MSG satellite over Reunion Island and Antananarivo Madagascar"

_EGUsphere, 2024_

## Referee Comment (RC1)

**Manuscript number:** egusphere-2024-1827

**Cloudiness retrieved from All-Sky camera and MSG satellite over Reunion Island and Antananarivo Madagascar** by Jean-Marcel Rivonirina et al.

**General Comments:**

This paper makes use of ground-based all-sky observation from the UV-Indien network provided by the Reuniwatt company. In order to measure cloud fraction, the camera images were analyzed using an existing industrial algorithm developed by the Reuniwatt company and another algorithm Elifan developed by CNRS. The authors have used all-sky images from two sites, Antananarivo, Madagascar, and Saint-Denis of Reunion Island, to evaluate the performance of two different algorithms used to analyze the all-sky images. Authors show strong agreement between the two algorithms Reuniwatt and Elifan (Bias=-5.48%, RMSE=6.48%, and r=0.99). To cover the large area authors used cloud products from the Meteosat Second Generation (MSG) satellites. Comparing camera and satellite observations a reasonable consistency between them has been shown, the significant biases are interpreted due to different spatial coverage by camera and satellite. Cloud properties in two locations in the southwest Indian Ocean, Antananarivo, Madagascar, and Saint-Denis of Reunion Island, have been examined using long term imager and MSG observations. Seasonal variabilities of cloud fraction over Antananarivo and Saint-Denis obtained from Reuniwatt and MSG during September 2019 - Jun 2022 has been reported. Antananarivo shows different diurnal variations in the cloud fraction during dry and wet seasons. However, in Saint-Denis, the morning skies are generally clear but cloud cover increases throughout the day during wet and dry seasons.

In summary, I see potential in the manuscript and it may eventually meet Annales Geophysicae's standards after a substantial **major revision** after addressing my concerns listed below.

**Specific comments:**

Lines 124-125: "In all cases, the sun pixels are considered as clear sky pixels in the segmented image". If the sun is in FOV, the pixels are expected to saturate in 30 sec. What are the saturation charges of the camera (CCD or CMOS) in ADU (Analog to Digital Unit)? Please clarify the meaning of levelling the pixels into the 'sun' category.

Lines 149-150: "To crop the observed image, we have selected a 70° radius angle around the zenith, which is equivalent to the angle chosen in the Reuniwatt algorithm" What is the logic to reducing FOV to 140º? As can be seen in Figure 2(a), in the white circle still building can be seen in the FOV.

Figure 3: Figure fonts are too small to be visible. The title of each image should be date in YYYY-MM-DAY and time in HH:MM:SS format with x and y axes values as the pixel number. I can see they are full images but do not have a size of 2kx2k; please clarify. How accurately do you generate the red circle in the images? What is the spatial resolution of the all-sky imagers used in this work?

Section 2.4.3 and Table 2: How the absolute threshold for cloud and blue sky was generated as shown in Table 3?

Figure 4: What are the advantages of giving pseudo color to the images? What is the unit of the scale shown in Figures 4b-e?

Section 2.6: what are xi and yi in the equations? Please mention them in the manuscript.

Lines 261-280: What is the area covered by MSG in 3x3 pixels vis-à-vis the area covered by a ground-based imager considered for this comparison? As correctly said "Increasing the window size tends to amplify the differences between camera and satellite observations." Therefore, before concluding that 3x3 pixels of MSG cover the same area, discussion on the area covered by the ground based observation needs to be compared with the MSG observations rather than trying different window sizes.

 **Technical corrections:**

Line 37: Please provide a reference for. "Generally, clouds cover more than half of the Earth's surface"

Line 71: every 15 mn should be every 15 minutes. You may use minutes or min throughout the manuscript.

Line 94 and 103: Generally, we follow (latitude, longitude) unless it becomes necessary to interchange them to convey a specific message. Be consistent in the manuscript text and Table 1 while mentioning lat, lon. How authors found a three-digit accuracy in lat. and long values?

Lines 100-101: "Our study contributes to filling the gap in our understanding of this site, allowing for inter-comparison of ground-based instruments with spatial observations" may be rephrased for clarity.

Line 114: Camera FOV is 180$^o$ for a fish-eye lens. What do authors mean by FOV of 360$^o$ x 180$^o$ around the site?

Lines 198-199:  correct "Bleu Sky" to "Blue Sky"

Figure 4: I cannot see the "red circle, and the object mask in black" in Fig. 4a.

Line 282: Figs. 7 and 8

---

## Author Comment (AC1)

**RESPONSE TO REFEREE 1 (RC1)**

First of all, we would like to thank the reviewer for dedicating his/her time to reading and providing feedback to improve the content of this work. Please find below the responses and explanations provided by authors (in regular font) to the reviewer's comments (in bold).

**RC1 : Lines 124-125: "In all cases, the sun pixels are considered as clear sky pixels in the segmented image". If the sun is in FOV, the pixels are expected to saturate in 30 sec. What are the saturation charges of the camera (CCD or CMOS) in ADU (Analog to Digital Unit)? Please clarify the meaning of levelling the pixels into the 'sun' category.**

Response 1 : Levelling the pixels into the "sun" category refers to how the image segmentation algorithm classifies the sun's pixels. Given the extreme brightness of the sun, the algorithm will identify and label those pixels as belonging to the "sun" category. This helps isolate the sun from other pixels and can be used to track or process the sun separately from the rest of the image.

Reuniwatt's camera relies on a (High Dynamic Range) HDR acquisition algorithm. The cloudy scene is captured multiple times with different exposure levels. The images are then combined to minimize as much as possible the saturated areas, especially around the sun. The exact exposition parameters are not provided.

**RC1 : Lines 149-150: "To crop the observed image, we have selected a 70° radius angle around the zenith, which is equivalent to the angle chosen in the Reuniwatt algorithm" What is the logic to reducing FOV to 140°? As can be seen in Figure 2(a), in the white circle still building can be seen in the FOV.**

Response 2 : A field of view (FOV) angle of 140° was selected to minimize the loss of information by eliminating only a small portion of the image, primarily containing the highest number of objects and pixels deformations at the horizon, while retaining the maximum amount of data on the cloud coverage present at the site. The limitation of removing certain objects led to the implementation of an additional processing step, specifically the creation of an object mask.

**RC1 : Figure 3: Figure fonts are too small to be visible. The title of each image should be date in YYYY-MM-DAY and time in HH:MM:SS format with x and y axes values as the pixel number. I can see they are full images but do not have a size of 2kx2k; please clarify. How accurately do you generate the red circle in the images? What is the spatial resolution of the all-sky imagers used in this work?**

Response 3 : The font size and number of images displayed in Figure 3 have been adjusted to make them more visible.

[Figure]

Figure 3. Simulated trajectory of the sun (red circle) on 29/08/2020, every hour from 4:00 to 12:00 (UTC).

To derive the equation for simulating the sun's trajectory in the image, we manually recorded samples of the sun's center positions in the images, under the condition that sun's center was visible. Subsequently, we performed equation fitting tests to optimize the determination of the most accurate sun position (red circle).

The raw images have a resolution of 2048 × 2048 pixels, while the preprocessed images, after cropping and excluding areas affected by pixel distortions near the horizon and object interferences around the site, have a resolution of 1271 × 1271 pixels.

**RC1 : Section 2.4.3 and Table 2: How the absolute threshold for cloud and blue sky was generated as shown in Table 3?**

Response 4 : The threshold values are visually identified through photo-interpretation of the pseudo-colored images (Fig. 4b), by comparing them with the real images (Fig. 4a). One of the main reasons for generating the pseudo-colored images is to facilitate this comparison.

**RC1 : Figure 4: What are the advantages of giving pseudo color to the images? What is the unit of the scale shown in Figures 4b-e?**

Response 5 : The advantages of giving pseudo color image have already been answered previously (Response 4).

The color scale in Figures 4b-e is dimensionless, as it represents the ratio of radiometric values (pixel values ranging from 0 to 255, corresponding to an 8-bit image) of the red and blue bands of the image (R/B).

Please find below the additional information.
"Figure 4. Intermediate image processing results at Moufia Saint-Denis - 30/09/2019 at 09:54:00 (UTC). (a) Cropped image containing the solar mask represented by the red circle, and the object mask in black. (b) Multi-color image processed using the absolute thresholding method. (c) Tricolor image processed with the absolute

thresholding method. (d) Multi-color image processed using the differential thresholding method. (e) Tricolor image processed using the differential thresholding method. The color scale (dimensionless) represents the ratio of radiometric values of the red and blue bands".

**RC1 : Section 2.6: what are xi and yi in the equations? Please mention them in the manuscript.**

Response 6 : As recommended by the referee, the following paragraph has been included. "xi and yi represent the algorithms that were compared (e.g., xi for Elifan and yi for Reuniwatt; or xi for Elifan or Reuniwatt and yi for MSG)".

**RC1 : Lines 261-280: What is the area covered by MSG in 3x3 pixels vis-à-vis the area covered by a ground-based imager considered for this comparison? As correctly said "Increasing the window size tends to amplify the differences between camera and satellite observations." Therefore, before concluding that 3x3 pixels of MSG cover the same area, discussion on the area covered by the ground based observation needs to be compared with the MSG observations rather than trying different window sizes.**

Response 7 : Regarding the field of view of the all-sky camera and the use of the 3x3 window, additional information has been provided. The observation radius of the camera depends on two parameters: the cloud height and the camera's field of view. In our case, it is primarily determined by the zenith angle of 70°, as represented by the formula below.

$$observation\ radius = cloud\ height \times \tan(70°) \approx 5,5\ km$$

Given that the cloud height over Réunion island is frequently observed at 2 km above surface level (Durand et al., 2021), and the 3x3 pixel window used by the MSG satellite covers a radius of approximately 4.5 km, this value is quite similar to the one obtained from ground-based observations.

Durand, J., Lees, E., Bousquet, O., Delanoë, J., Bonnardot, F. Cloud Radar Observations of Diurnal and Seasonal Cloudiness over Reunion Island. Atmosphere, 12, 868. https://doi.org/10.3390/atmos12070868, 2021

**RC1 : Line 37: Please provide a reference for. "Generally, clouds cover more than half of the Earth's surface"**

Response 8 : The reference (Liu et al., 2023) has been added to the paper.

Liu, H., Koren, I., Altaratz, O., and Chekroun, M. D.: Opposing trends of cloud coverage over land and ocean under global warming, Atmos. Chem. Phys., 23, 6559–6569, https://doi.org/10.5194/acp-23-6559-2023, 2023

**RC1 : Line 71: every 15 mn should be every 15 minutes. You may use minutes or min throughout the manuscript.**

Response 9 : The notation « 15 mn » has been replaced by « 15 min ».

**RC1 : Line 94 and 103: Generally, we follow (latitude, longitude) unless it becomes necessary to interchange them to convey a specific message. Be consistent in the manuscript text and Table 1 while mentioning lat, lon. How authors found a three-digit accuracy in lat. and long values?**

Response 10 : Thank you for your valuable observation. The notation « lon, lat » will be swapped to "lat, lon". The assignment of coordinates with precision of three digits is not adequate; therefore, we have reduced the precision to two digits.

**RC1 : Lines 100-101: "Our study contributes to filling the gap in our understanding of this site, allowing for inter-comparison of ground-based instruments with spatial observations" may be rephrased for clarity.**

Response 11 : Please find the improved version below.
"Our study contributes to advancing the understanding of this site by facilitating the comparative analysis of ground-based instruments and spatial observations".

**RC1 : Line 114: Camera FOV is 180° for a fish-eye lens. What do authors mean by FOV of 360° x 180° around the site?**

Response 12 : The sentence has been rephrased to avoid misunderstanding. "360° x 180° represents spherical view angle of the camera".

**RC1 : Lines 198-199: correct "Bleu Sky" to "Blue Sky"**

Response 13 : « Bleu Sky » has been corrected to « Blue Sky ».

**RC1 : Figure 4: I cannot see the "red circle, and the object mask in black" in Fig. 4a.**

Response 14 : To ensure better coherence in the order of figure presentation, we have changed the notation from Fig. 4 to Fig. 3 (line 155). Additionally, the quality of the latter has been enhanced to improve the visibility of the masks (sun and object).

**RC1 : Line 282: Figs. 7 and 8**

Response 15 : Figs. 7 and 8 have been considered in the paper.

---

## Author Comment (AC2)

**RESPONSE TO REFEREE 2 (RC2)**

We would like to thank the reviewer for taking the time to read and provide valuable comments, which have helped us improve the content of this work. The author's responses are written in regular font, while the reviewer's comments are in bold.

**RC2 : It compares different data sources, but remains very general in introduction and state of the art, describes the used data only roughly, and does not discuss the restrictions of the chosen methods sufficiently.**

Response 1 : We thank the reviewer for his/her comment; indeed, we should have discussed potential limitations. "Since we are comparing two different observation methods, the differences between the measurements are apparent. These differences are generally attributed to the distinct observation positions of the two systems. The camera can be significantly influenced by low clouds, whereas satellites primarily observe clouds at higher levels, especially when all three cloud layers (low, medium, and high) are present. These discrepancies may also partly result from variations in the field of view of the two observation methods".

These remarks have been considered, and restructuring has been carried out to improve the manuscript.

**RC2 : It should be better elaborated what the new data or new research results are compared to existing knowledge.**

Response 2 : Although various climate measurement instruments exist, information on cloud cover in the South West Indian Ocean (SWIO) remains limited due to the lack of dedicated instruments. The installation of cloud fraction measurement stations from the UV-Indien network at these sites will help enrich current databases. The SWIO regions are particularly vulnerable to the effects of climate change or natural disaster, which impact the population annually (Leroux et al., 2024). This study opens new perspectives for monitoring the evolution of climate variability in each of these sites.

All of the motioned highlights of new data and its importance are added to the manuscript; in section 3.1.2 Camera against Satellite.

Leroux, M. D., F. Bonnardot, S. Somot, et al.: "Developing Climate Services for Vulnerable Islands in the Southwest Indian Ocean: A Combined Statistical and Dynamical CMIP6 Downscaling Approach for Climate Change Assessment." Climate Services 34: 100491, https://doi.org/10.1016/j.cliser.2024.100491, 2024

**RC2 : Overall, the study is very general, seems to be of local interest mainly, and does not clearly add new knowledge to the understanding of cloud climatologies beyond what is known from general meteorology and existing satellite or model-based climatologies.**

Response 3 : This study provides a new understanding of climatology in this region and beyond, by analyzing a more precise timescale with higher spatial resolution compared to the currently available data. The enhanced resolution could benefit the scientific and application communities, such as in short-term solar resource prediction.

**RC2 : The abstract is very unclear, it provides numbers on bias etc. but it is not clear for which geophysical parameter, in general the abstract does not tell which parameter is analysed and seems to repeat twice statistics for Saint-Denis? At the end the reader may guess that it is about cloud coverage? Later in line 56 it is called cloud fraction?**

Response 4 : The abstract has been rewritten, emphasizing the cloud fraction as the primary parameter under study.

**RC2 : The commercial algorithm of Reuniwatt is not further specified by the coauthors originating partly from the company Reuniwatt. This is a drawback in a scientific paper which only can be accepted if the method itself is not relevant and the other results of the paper are standing on its own.**

Response 5 : Although the description of the Reuniwatt algorithm is considered insufficient, the effectiveness of its application has already been demonstrated in several previous publications (Cadet et al., 2020; Lamy et al., 2021). Furthermore, the relevance of its results has been confirmed through a comparison with the Elifan algorithm.

Cadet, J-M., Portafaix, T., Bencherif, H., Lamy, K., Brogniez, C., Auriol, F., Metzger, J-M., Boudreault, L-E., Wright, C.Y.: Inter-Comparison Campaign of Solar UVR Instruments under Clear Sky Conditions at Reunion Island (21°S, 55°E). International Journal of Environmental Research and Public Health. 17(8):2867. https://doi.org/10.3390/ijerph17082867, 2020.

Lamy, K.; Ranaivombola, M.; Bencherif, H.; Portafaix, T.; Toihir, M.A.; Lakkala, K.; Arola, A.; Kujanpää, J.; Pitkänen, M.R.A.; Cadet, J.-M. Monitoring Solar Radiation UV Exposure in the Comoros. Int. J. Environ. Res. Public Health, 18, 10475. https://doi.org/10.3390/ijerph181910475, 2021.

**RC2 : The Elifan algorithm on the other hand is further detailed, but I miss a clear description why it is better/different from other existing cloud masking methods for all-sky cameras.**

Response 6 : The application of various mask types (sun and object) to enhance measurement accuracy is one of the key strengths of this algorithm. Furthermore, the use of two distinct thresholding methods (absolute and differential) improves the algorithm's precision compared to other image processing algorithms.

**RC2 : I'm missing a specific review of the state of the art in cloud masking of all-sky cameras in the introduction section. Why is the presented work going beyond the state of the art? What is new? Why is the comparison of the two camera algorithm of interest for the reader?**

Response 7 : The introduction has been revised with focus on the main objective of the study, while incorporating additional relevant information. This revision highlights the recent advancements and existing gaps in the current understanding of the use of satellites and all-sky cameras. Indeed, the use of these technologies has significantly evolved in recent years, allowing for improved observation of atmospheric and spatial phenomena. However, despite these advancements, several unresolved issues remain, particularly regarding the accuracy of measurements and the enhancement of global coverage. This study aims to address these gaps by exploring new application approaches for these technologies, thereby contributing to the further development of knowledge in this field.

**RC2 : Also, the introduction is very broad and not specific enough to describe the research question of this paper clearly. It should be rewritten with focus on the study details and not refer generally to textbook knowledge.**

Response 8 : As previously mentioned, the paper will be reorganized and rewritten, particularly in the introduction and methodology sections, in order to ensure better coherence and logical flow.

**RC2 : It should be better justified why the assessment of the two locations is of special interest to the broader scientific community. What is known on cloudiness in the area from other studies? How does the work fit to the existing state of the art? Just the aim to understand the cloud properties of two locations operated by the author team is of minor interest to the scientific community.**

Response 9 : Cloud variability in the south-western Indian Ocean is linked to the vast oceanic coverage, which fosters the development of tropical cyclones. It also influences UV radiation: periods of clear sky increase health risks, while fragmented clouds can amplify UV exposure. This variability has major societal impacts, particularly through cyclones and UV exposure. From a scientific perspective, studying this region is crucial to understanding the interactions between cloud cover and extreme climatic phenomena. The use of ground-based and satellite databases is essential for improving the accuracy of climate models and forecasts. Our study focuses on Antananarivo (Madagascar) and Réunion, two representative and strategic areas of the south-western Indian Ocean, in order to better understand the climatic dynamics of the region.

**RC2 : The very general, again textbook like satellite description could be replaced by a much better description on the details of cloud mask retrieval and how the assumptions and restrictions of the CLM algorithm may affect your study. What is known from other studies on the CLM product accuracy?**

Response 10 : The manuscript will be reorganized so that the satellite cloud mask retrieval is more detailed in the description section rather than in the results section. Additional information, including the limitations and restrictions that could affect the algorithm's quality, will be added to the manuscript.

**RC2 : How is cloud fraction inside a satellite pixel handled? The authors seem to assume that the CLM product provides only 0 and 100% cloud fraction inside a pixel? Is this realistic? What is the impact of this assumption? What happens to inside-a-pixel cloud fraction and what happens to sub-visible clouds which should be visible in the all-sky cameras, but not in the satellite images?**

Response 11 : It is important to note that CLM and cloud fraction are two distinct variables. CLM, in binary format, only indicates whether a pixel represents a cloud or not. In contrast, the cloud fraction is derived through an additional calculation applied to the CLM, yielding a value that represents a range of measurements from 0 to 100%.

$$\text{cloud fraction} = \frac{cloud\ pixel\ number}{total\ pixel\ number}$$

This information will be added to the algorithm description section to better explain how the cloud fraction was derived.

**RC2 : The reprojection of satellite-based cloud masks by a 3x3 window is assumed to be the same area as the field of view of all-sky cameras. The actual field of view of the all-sky camera depends very much on visibility and the type and height of clouds in the field of view. It can range from very local to hundreds of km. This is not taken into account. This is a major drawback of the study.**

Response 12 : Regarding the field of view of the all-sky camera and the use of the 3x3 window, additional information has been provided. The observation radius of the camera depends on two parameters: the cloud height and the camera's field of view. In our case, it is primarily determined by the zenith angle of 70°, as represented by the formula below.

$$observation\ radius = cloud\ height \times \tan(70°) \approx 5{,}5\ \text{km}$$

Given that the cloud height over Réunion island is frequently observed at 2 km above surface level (Durand et al., 2021), and the 3x3 pixel window used by the MSG satellite covers a radius of approximately 4.5 km, this value is quite similar to the one obtained from ground-based observations.

Durand, J., Lees, E., Bousquet, O., Delanoë, J., Bonnardot, F. Cloud Radar Observations of Diurnal and Seasonal Cloudiness over Reunion Island. Atmosphere, 12, 868. https://doi.org/10.3390/atmos12070868, 2021

**RC2 : What is the additional value of using all-sky cameras for cloud fraction quantification if the satellite data is already available as open data? If you compare only against satellite data, what is the added value of an all-sky camera? The research question of the study is not well discussed and analysed. The relevance of the work should be better elaborated.**

Response 13 : Although satellite measurements are freely available, all-sky cameras provide data at a finer spatial and temporal scale. Moreover, their storage cost is significantly lower compared to satellite images. They are also essential for filling data gaps in case of technical issues with satellites.

**RC2 : Why do you introduce cloud type? It is unclear how this is used in the study.**

Response 14 : Cloud fraction offers a broader view of cloud variability, but it may sometimes miss specific details that can be clarified through cloud type. Cloud type provides essential additional context for interpreting certain

cloud characteristics, offering a deeper understanding of variability that cannot be fully captured by cloud fraction alone.

**RC2 : What is meant by the term 'nebulosity'?**

Response 15 : "Nebulosity" refers to the amount of clouds present in the sky at a given moment, or more precisely, the proportion of the sky covered by clouds. It is often expressed in oktas or as a percentage, where 0/8 (or 0%) represents a completely clear sky, and 8/8 (or 100%) indicates a sky fully covered with clouds. The intermediate values corresponding to partial cloud cover.

**RC2 : What is the difference between cloud fraction, cloud coverage, cloud fraction index? Please better define the terminology used and be more specific.**

Response 16 : You are right that the three terms may cause some ambiguity. However, we have chosen to use the term "cloud fraction," as it is the predominant parameter in the literature and the one we aim to highlight in this study.

**RC2 : I'm not going into more technical details of the paper as it first requires a significant rewriting of the overall study design.**

Response 17 : As previously mentioned, the paper will be reorganized and rewritten, particularly in the introduction and methodology sections, to ensure better coherence and logical flow.